# Progression and Dissemination of Pulmonary Mycobacterium Avium Infection in a Susceptible Immunocompetent Mouse Model

**DOI:** 10.3390/ijms23115999

**Published:** 2022-05-26

**Authors:** Raymond Rosenbloom, Igor Gavrish, Anna E. Tseng, Kerstin Seidel, Shivraj M. Yabaji, Hans P. Gertje, Bertrand R. Huber, Igor Kramnik, Nicholas A. Crossland

**Affiliations:** 1Graduate Medical Sciences, Boston University School of Medicine, Boston, MA 02118, USA; rayrosen@bu.edu; 2National Emerging Infectious Diseases Laboratories, Boston University, Boston, MA 02215, USA; igorgavr@bu.edu (I.G.); annatsng@gmail.com (A.E.T.); kseidel@bu.edu (K.S.); shivrajyabaji@gmail.com (S.M.Y.); hgertje@bu.edu (H.P.G.); 3Department of Pathology & Laboratory Medicine, Boston University School of Medicine, Boston, MA 02118, USA; 4Department of Neurology, Boston University School of Medicine, Boston, MA 02118, USA; huberb@bu.edu; 5Department of Medicine, Boston University School of Medicine, Boston, MA 02118, USA

**Keywords:** nontuberculosis mycobacteria, granulomatous pneumonia, macrophage polarization, digital image analysis, histopathology

## Abstract

Pulmonary infections caused by the group of nontuberculosis mycobacteria (NTM), Mycobacterium avium complex (MAC), are a growing public health concern with incidence and mortality steadily increasing globally. Granulomatous inflammation is the hallmark of MAC lung infection, yet reliable correlates of disease progression, susceptibility, and resolution are poorly defined. Unlike widely used inbred mouse strains, mice that carry the mutant allele at the genetic locus sst1 develop human-like pulmonary tuberculosis featuring well-organized caseating granulomas. We characterized pulmonary temporospatial outcomes of intranasal and left intrabronchial *M. avium* spp. hominissuis (M.av) induced pneumonia in B6.Sst1S mice, which carries the sst1 mutant allele. We utilized traditional semi-quantitative histomorphological evaluation, in combination with fluorescent multiplex immunohistochemistry (fmIHC), whole slide imaging, and quantitative digital image analysis. Followingintrabronchiolar infection with the laboratory M.av strain 101, the B6.Sst1S pulmonary lesions progressed 12–16 weeks post infection (wpi), with plateauing and/or resolving disease by 21 wpi. Caseating granulomas were not observed during the study. Disease progression from 12–16 wpi was associated with increased acid-fast bacilli, area of secondary granulomatous pneumonia lesions, and Arg1+ and double positive iNOS+/Arg1+ macrophages. Compared to B6 WT, at 16 wpi, B6.Sst1S lungs exhibited an increased area of acid-fast bacilli, larger secondary lesions with greater Arg1+ and double positive iNOS+/Arg1+ macrophages, and reduced T cell density. This morphomolecular analysis of histologic correlates of disease progression in B6.Sst1S could serve as a platform for assessment of medical countermeasures against NTM infection.

## 1. Introduction

The nontuberculosis mycobacteria (NTM) of Mycobacterium avium complex (MAC) causes pulmonary disease in humans with increasing prevalence worldwide partly attributed to climate change and increased density of humans within tropical regions, where NTM are most prevalent [1,2]. MAC consists of at least nine environmental organisms, with Mycobacterium avium (M.av) being among the most common respiratory isolates [3]. While the increasing burden of MAC pulmonary disease is mostly documented in high-income countries, MAC prevalence may be underappreciated and misdiagnosed with Tuberculosis (TB) in settings with limited diagnostic capabilities [4].

MAC infection in humans usually causes either disseminated miliary interstitial or bronchogenic pulmonary disease. Disseminated MAC infection usually occurs in the setting of the Acquired Immune Deficiency Syndrome (AIDS) and was associated with a short median survival of 103–134 days prior to the development of antiretroviral therapy [5]. Disseminated MAC is also associated with the Mendelian Susceptibility to Mycobacterial Disease [6], further emphasizing the essential roles of CD4+ T cells, and the IL-12—Interferon gamma (IFNγ) axis in anti-mycobacterial immunity. Paralleling the immune response in humans, CD4+ Th1 cells producing IFNγ primarily mediate the adaptive immune response to NTM infection [7,8,9].

Bronchogenic MAC disease is strongly associated with structural lung defects caused by chronic obstructive pulmonary disease and cystic fibrosis, systemic autoimmune rheumatic diseases, use of immunosuppressive drugs such as TNF-alpha blockers, and malnutrition [6]. Pulmonary MAC is also associated with >60 years of age [10]. In more than half of untreated human patients diagnosed with bronchogenic MAC infection, the disease is progressive leading to bronchiectasis with cavitary formation and necrotizing bronchocentric granulomatous inflammation [11]. This disease type is particularly unresponsive to antibiotic treatment, requiring an 18–24-month multidrug regimen with high toxicity [4,12]. Mechanisms of susceptibility to pulmonary MAC infection, however, are multifactorial and incompletely understood [6,10].

The histologic hallmarks of MAC pulmonary infection are granulomas or granulomatous infiltrate with the presence of intralesional acid-fast bacilli (AFB) [6]. The radiographic presentation of MAC pulmonary disease is usually fibrocavitary or characterized by nodules and bronchiectasis [6]. MAC-positive granulomas with areas of caseous necrosis, epithelioid and multinucleated CD68+ giant cells have been described in the lungs of immunocompetent patients with bronchiectatic and cavitary forms of disease [13,14]. The signals that drive mycobacterial granuloma formation have largely been defined as mediated by the Th1 immune response consisting of IFNγ and classical macrophage activation (M1) that mediate host resistance. However, alternatively activated macrophages (M2) that express Arginase-1 (Arg1) typically associated with the Th2 driven immune response have also been described in TB and NTM granulomas [15,16], where they may play immunomodulatory or pro-fibrotic roles [16,17]. Recently, type 2 macrophage responses were shown to contribute to the necrotization of mycobacterial granulomas in a zebrafish model [18]. Our current understanding of mechanisms that underlie macrophage polarization in mycobacterial granulomatous infiltrate, as well as its roles in the necrotization and organization of mycobacterial granulomas, is limited [19].

Because the structure of granulomas influences whether pharmaceutical compounds or a vaccine-induced immune response can penetrate granulomas to eradicate mycobacteria [20,21,22], the development of predictive mouse models that mimic the diversity of human pulmonary NTM disease is particularly important. These models would be especially valuable for studying mechanisms underlying lung tissue damage and testing combinations of antimicrobials, host-directed therapies, and vaccines. Experimental M.av infections in immunocompetent mice has demonstrated that the genetic background of the host, route and dose of infection and M.av strains, laboratory, or recent clinical isolates, all play important roles in establishing sustainable infection and determining the disease severity outcomes [22,23,24].

Prior work has shown that the resistant allele of Nramp1 confers protection against aerosol infection with M.av (strain 724) [25]. The NRAMP1 gene encodes the proton-coupled divalent metal ion transporter SLC11A1 that limits iron availability to bacterial pathogens that are localized inside phagocytic vacuoles and thus reduce their intravacuolar replication [26]. Human ortholog(s) of this gene are implicated in several mycobacterial diseases, including Buruli ulcer and TB [27,28]. However, recent work using the C3HeB/FeJ mouse strain that carries the Nramp1 resistant allele demonstrated that aerosol infection with M.av (strain 2285) resulted in chronic progressive lung infection and the formation of necrotic granulomas unlike those seen in other standard inbred mouse strains [22], indicating that the NRAMP1 mediated resistance mechanism is not sufficient to prevent the necrotization of M.av granulomas in that model. These inbred C3HeB/FeJ mice are immunocompetent but highly susceptible to virulent M. tb. This unique necrotic granuloma phenotype is controlled by a single genetic locus sst1 (supersusceptibility to tuberculosis-1), which has been characterized in our previous studies [29,30,31,32,33]. This locus controls macrophage responses to Tumor necrosis factor (TNF) and the sst1 susceptibility allele reduces macrophage stress resilience and increases type I interferon production [34,35,36,37]. We decided to use the mouse strain B6.Sst1S because it carries three susceptibility alleles identified previously. The sst1 susceptible allele was shown to control the extent of lung pathology during infection with virulent M. tb [31,38], Chlamydia pneumoniae [36], and Legionella pneumophila [39]. This mouse strain also carries the susceptibility allele of the Scl11a1 gene that has been identified as a major genetic factor controlling replication of taxonomically unrelated intracellular pathogens including M.av [40]. It also carries the MHC class II allele b of the H2-Aβ that has been associated with increased lung pathology in M.av-infected mice [41]. Therefore, we postulated that the genetic composition of the B6.Sst1S mice may provide a model of pulmonary NTM infections that better resembles destructive pulmonary NTM infections in immunocompetent, but susceptible humans.

The goal of this study was to determine if we could induce M.av pulmonary lesions that recapitulate human clinical phenotypes (bronchogenic and/or disseminated) in B6.Sst1S congenic mice and to determine if they develop divergent morphomolecular signatures compared to B6 wild-type (WT) mice. B6.Sst1S mice exhibited large secondary granulomatous lesions in un-inoculated lung lobes, with peaked bacterial loads at 16 wpi. Macrophage polarization within pulmonary granulomas was further characterized using fluorescent multiplex immunohistochemistry (fmIHC) and quantitative image analysis (IA). Furthermore, larger secondary granulomatous lesions contained increased double positive iNOS/Arg1 macrophages during high bacterial timepoints (16 wpi) in the un-inoculated lung lobes. In comparison to B6 WT mice, the lungs of B6.Sst1S mice had greater area of mycobacterial antigen immunoreactivity and AFB, larger secondary granulomatous lesions in the un-inoculated lobes, increased double positive iNOS/Arg1 macrophages, and reduced CD3ε+ T cell density.

In summary, we were able to confirm a more severe disseminated lung pathology in B6.Sst1S mice compared to B6 WT mice through development of an intrabronchiolar model of *M. avium* that peaked around 16 wpi that exhibited both bronchogenic (inoculated lung lobe) and disseminated interstitial disease (un-inoculated lung lobes). By utilizing fmIHC and quantitative IA, we identified distinct molecular signatures of disease progression and plateauing/resolution. The B6.Sst1S intrabronchiolar *M. avium* mouse model will be a useful tool for preclinical evaluation of therapeutics and candidate vaccines against M.av infection for disseminated phenotypes [42]. More virulent clinical isolates would need to be evaluated to determine if bronchiectasis and necrotizing cavitary disease could be produced to better recapitulate this human clinical outcome more faithfully.

## 2. Results

### 2.1. Histopathological Features of Lung Lesions Due to Intranasal Inoculation in B6.Sst1S Mice

To determine whether chronic pulmonary disease can be induced in B6.Sst1S mice with Mycobacterium avium strain 101 (M.av), 10^6^ of M.av was delivered intranasally. There was progressive growth of mycobacteria in the lungs of the B6.Sst1S mice between 4- and 8–10-weeks post infection (wpi) and plateauing of bacterial loads by 12 wpi (Figure 1A). There were statistically significant increases in the lung histology ordinal scores at each timepoint (8, 10, 12 wpi) compared to 4 wpi (*p* = 0.0127, *p* = 0.0009, *p* ≤ 0.0001 respectively) (Figure 1B).

Microscopically, pulmonary lesions at 4 wpi were represented primarily by minimal focal-to-multifocal peribronchiolar and perivascular lymphohistiocytic cuffing with minimal interstitial granulomatous infiltrate (presence of epithelioid macrophages and/or rare multinucleated giant cells) affecting less than 10% of lung parenchyma. By 8 wpi, the lesions progressed to mild-to-moderate multifocal lymphohistiocytic and granulomatous inflammation that occupied 10–30% of the lung parenchyma. A board-certified pathologist (N.A.C.) utilized a semi-quantitative ordinal scoring system to assess lung pathology from each specimen (Appendix A). At 12 wpi, mean histopathology scores peaked, represented by moderate-to-marked multifocal lymphohistiocytic and granulomatous inflammation that affected >30–50% of lung parenchyma (Figure 1C). Acid fast bacilli (AFB) staining (Figure 1D) and IHC staining with antibodies targeting mycobacterial antigens (Figure 1E) demonstrated intra-histiocytic and cytoplasmic mycobacteria within areas of granulomatous pneumonia. However, neither extracellular bacteria, nor macro-necrosis or organized necrotic granulomas were observed. No apparent difference in severity was evident in left vs. right lung lobes at any time point examined.

### 2.2. Histopathological Features of Lung Lesions in Inoculated (Left) and Un-Inoculated Lung Lobes (Right) Due to Unilateral Intrabronchial Inoculation

Tissue sections were analyzed at 12, 16, and 21 wpi. The lesions in the inoculated lung lobe (left) were characterized by extensive peribronchiolar and interstitial lymphohistiocytic and granulomatous pneumonia (Figure 2A). The un-inoculated lung lobes (right) possessed prominent granulomatous lesions at each time point examined; however, these lesions displayed a more random interstitial distribution with no direct affiliation with bronchioles (Figure 2B). At each time point examined (12, 16, 21 wpi), ordinal histopathology scores tended to be increased in the inoculated compared to un-inoculated lung lobes attributed to the larger and coalescing nature of the lesions (Figure 2C, Appendix A).

To further characterize the myeloid and lymphoid cell populations of classified granulomatous lesions, fluorescent multiplex immunohistochemistry (fmIHC) was employed to phenotype macrophage populations using the markers CD68, iNOS, and Arg1, and T cells with CD3ε. We utilized a random forest Tissue Classifier (TC) for automated identification and annotation of pulmonary granulomatous lesions for subsequent downstream analysis (see Methods). Granulomatous lesions in inoculated-lobe lesions encompassed a great total area of lung parenchyma and were more continuous and less discrete (Figure 2D) compared to smaller well demarcated interstitial secondary lesions in the un-inoculated lung lobes (Figure 2E).

Analysis of cell populations within granulomatous lesions identified further differences between the inoculated and un-inoculated lung lobes. Inoculated lobe lesions qualitatively had higher intensity M.av immunoreactivity within iNOS+ macrophages (Figure 2D, lower) compared to lesions within the un-inoculated lobes. There were also large areas of bronchial associated lymphatic tissue (BALT) as evidenced by areas of high density DAPI staining with intralesional CD3ε+ T cells adjacent to bronchioles (Figure 2D). Within lesions of the un-inoculated lobes, there was decreased area and qualitative intensity of M.av immunoreactivity (Figure 2E). The secondary lesions within un-inoculated lobes were not associated with bronchioles; however, lymphoid aggregates were occasionally observed to be affiliated with these lesions (Figure 2E).

Inoculated lung lobes tended to have larger lesion area (*p* value = 0.0091) and perimeter (*p* value = 0.0031) than un-inoculated lung lobe lesions at 16 wpi (Figure 2F,G). M.av AFB area and T cell density (*p* value = 0.0464) were also greater in inoculated lobe lesions compared to lesions in the un-inoculated lung lobes at 16 wpi (Figure 2H,I). Increased mean lesion area, M.av AFB, and T cell immunoreactivity were consistently increased in the inoculated lung lobes at each time point examined (Appendix A).

Taken together, histopathology and fmIHC demonstrated a common trend in which lesions in the inoculated lobes could be distinguished from lesions in the un-inoculated lobes by enhanced pathology, association with bronchioles, increased mycobacterial immunoreactivity and greater T cell density. Given the unique features observed between the lung lobes, subsequent analysis focused on defining immune signatures of discrete lesions in the un-inoculated lobes, which were classified as secondary lesions using criteria defined in the Methods.

### 2.3. Progression of Secondary Lesions in the Un-Inoculated Lobes (Right) after Left Intrabronchial Inoculation

After observing disparate histopathological and immune cell infiltrates between inoculated and un-inoculated lobes, further analysis was conducted to study the temporal progression of secondary lesions at 12, 16, and 21 wpi (Figure 3). At 12 wpi, secondary lesions were characterized by loosely cellular granulomatous pneumonia (Figure 3A). By 16 wpi, alveolar macrophages displayed altered cytoplasmic morphology over time suggestive of a reactive phenotype represented by an increased area of lacy eosinophilic cytoplasm, with a greater overall total percentage of lung involvement (Figure 3B). Additionally, necrosis and identification of multinucleated macrophages became more frequent at later time points (Figure 3B). A trend toward increased ordinal histopathology scores was observed at later time points compared to 12 wpi (Figure 3C). In addition to histopathological analysis, fmIHC and image analysis were performed to characterize cell types within granulomatous lesions.

Analysis of cell populations within secondary lesions of advanced lesions were characterized by distinctive biomarker patterns. At 12 wpi, secondary lesions contained iNOS+ and iNOS− macrophages with few, sporadic Arg1+ macrophages (Figure 3D). There was punctate M.av immunoreactivity in some but not all iNOS+ macrophages (Figure 3D, upper). T cells were haphazardly distributed in the lesion, as well as congregated together in tertiary lymphoid aggregates (Figure 3D, upper). At 16 wpi, a greater proportion of macrophages were iNOS+ and there was a greater presence of Arg1+ macrophages, including double positive iNOS+/Arg1+ macrophages (Figure 3E). In comparison to 12 wpi, these iNOS+ and double positive macrophages appeared more reactive with greater cytoplasmic area (Figure 3E). Additionally, M.av immunoreactivity was brighter and comprised a larger area at 16 wpi compared to 12 wpi (Figure 3E, upper). Quantitative analysis demonstrated a temporal progression of increased mean granulomatous lesion area (12–21 wpi *p* value = 0.0488, 12–21 wpi *p* value = 0.0019), and percentage area of iNOS and CD3ε+ T cells across the secondary lesion-containing un-inoculated lung lobes at 12, 16, and 21 wpi. (Figure 3F–H).

AFB staining and quantification was also employed on these specimens in both the inoculated and un-inoculated lung lobes to visualize and quantify intact bacteria (Appendix A). In both lobes, AFB peaked at 16 wpi and then declined by 21 wpi (Appendix A). Qualitatively, both 16 and 21 wpi lesions were characterized by dense cytoplasmic bacterial aggregates in comparison to 12 wpi, where bacteria more often were individualized and of sparse number (Appendix A). These findings in combination with histopathology and lesion size comparisons indicate that bacterial loads peaked at 16 wpi; however, numerous residual bacilli and granulomatous lesions remained at 21 wpi.

Secondary lesions at 12 and 16 wpi were subsequently classified as “small” or “large,” as previously described in the Methods. Size stratification analysis facilitated the identification of divergent macrophage subpopulations in variably sized granulomatous lesions.

At 12 wpi, small lesions were characterized by clusters of iNOS+ CD68+ macrophages with sparse intracellular M.av immunoreactivity and interspersed T cells (Appendix A). Large lesions had similar cellular populations as small lesions at 12 wpi (Appendix A). Quantitative analysis comparing percentage area of various fluorescent overlap of iNOS/Arg1/M.av reactivity did not reveal differences between lesions (Appendix A). There was no overlap of Arg1+/Inos−/M.av+ and Arg1+/iNOS+/M.av + areas in small or large secondary lesions at 12 wpi. This finding suggests that the infected Arg1 macrophage phenotype was not a dominant feature of this earlier time point.

At 16 wpi, large secondary lesions contained iNOS+ and Arg1+ macrophages with large cytoplasmic area (Figure 4A). Arg1+ macrophages were generally found in central parts of these large lesions (Figure 4A). There were also areas of high density DAPI staining with scattered CD3ε+ cells, which are interpreted as tertiary lymphocytic aggregates (Figure 4A). A relatively smaller proportion of reactive macrophages, as indicated by larger cytoplasmic areas found in large secondary lesions, were Arg1+ (Figure 4A, second column). These Arg1+ macrophages were frequently but not exclusively iNOS+ and mostly contained punctate M.av immunoreactivity (Figure 4A). A larger proportion of macrophages were iNOS+ (Figure 4A, third column). iNOS+ macrophages were uniformly CD68+ and contained M.av immunoreactivity (Figure 4A, fourth column). In contrast, small secondary lesions tended to have macrophages with smaller cytoplasmic iNOS staining (Figure 4B). Arg1+ macrophages were generally not present and there were few T cells interspersed within the lesions (Figure 4B).

Quantitative analysis of all secondary lesions revealed differential expression patterns of iNOS and Arg1 expressing macrophages. iNOS was more commonly associated with infection than Arg1+ macrophages. In small lesions, there was a greater percentage of iNOS macrophage immunoreactivity overlapped with M.av (45.75%) compared to Arg1 M.av overlap (14.43%) (*p* ≤ 0.0001) (Figure 4C). This trend was the same in large lesions, with greater percentage iNOS M.av overlap (46.00%) compared to Arg1 M.av overlap (18.56%) (*p* ≤ 0.0001) (Figure 4D). iNOS was strongly positively correlated with mycobacterial area (R squared = 0.8082; *p* ≤ 0.0001), whereas the correlation between Arg1+ and M.av was lower (R squared = 0.5941; *p* ≤ 0.0001) (Figure 4E). Next, the area of iNOS and Arg1 immunoreactivity was compared directly between small and large lesions at 16 wpi. Percentage iNOS+ area was found to be greater in small compared to large lesions (Figure 4F) (*p* = 0.008). Conversely, area of Arg1+ and double positive iNOS+/Arg1+ macrophages tended to be greater in large lesions (*p* = 0.0097) (Figure 4G–I). These findings are consistent with qualitative observations previously described (Figure 4A,B) showing increased iNOS+ and/or Arg1+ positive macrophages in large secondary lesions. Taken together, these data demonstrate that iNOS was a dominant infected cell phenotype in all small and large secondary lesions and thereby was strongly positively correlated with M.av infection.

After characterizing these macrophage phenotype patterns associated with lesion progression, identifying whether various phenotypes of myeloid and non-myeloid cells expressed IFNβ was a priority. To determine the identity and distribution of IFNβ expressing cells, the B6.Sst1S, IFNβ-YFP reporter mouse in which YFP serves as a surrogate for IFNβ expression was employed and analyzed at 23 wpi following M.av infection. To analyze representative lesions, an fmIHC cocktail that included GFP specific antibodies, which served as the surrogate marker for IFNβ, was utilized. Qualitative analysis of fmIHC demonstrated that IFNβ predominantly colocalized with iNOS+ M.av+ macrophages (Figure 5A). Additionally, IFNβ+ (iNOS−/Arg1−) cells with diffuse cytoplasmic expression were found within perivascular mononuclear infiltrate and the pulmonary interstitium (Figure 5B). IFNβ+ cells were not typically found in T cell-rich lymphocytic aggregates (Figure 5B). There was no observable co-expression of Arg1+ macrophages and IFNβ in these lesions (Figure 5A,B). Overall, IFNβ expression was predominantly found in M.av infected macrophages, which were most commonly iNOS+. These findings were confirmed via serial sections using single marker staining with Mycobacteria- and GFP-specific antibodies (Figure 5C,D).

### 2.4. The sst1-Mediated Susceptibility to M. avium Infection

To investigate the specific effect of the sst1 locus in this model, progression of pulmonary M.av infection in wild type B6 and B6.Sst1S mice at 12 and 16 wpi was assessed. Ordinal histopathology scores obtained using H&E staining did not differ between WT and susceptible mice at individual timepoints (Appendix A). Next, quantitative whole lung analysis was used to compare lung pathology between mouse strains. No differences in lesion area and M.av immunoreactivity were found at 12 wpi (data not shown). At 16 wpi, there were no statistically significant differences in lesion area in the inoculated lung lobes (Figure 6A). However, there was a statistically significant decrease in percentage mean lesion area (*p* value = 0.0373) in the sst1 resistant B6 WT un-inoculated lung lobes (Figure 6B). AFB staining demonstrated decreased bacterial counts in B6 WT inoculated (*p* = 0.0155) and un-inoculated (*p* = 0.0413) lung lobes at 16 wpi (Figure 6C,D). These findings suggest that the Sst1 mechanism of resistance was exhibited by 16 wpi with characteristic larger secondary lesions in the un-inoculated lung lobes.

Therefore, we compared the differential cell populations in secondary lesions of B6.Sst1S and B6 WT mice at 16 wpi using fmIHC and AQ analysis. For these analyses, lesions were stratified by size, as discussed previously. Consistent with the AFB analysis, less M.av immunoreactivity was also qualitatively observed in both large and small B6 WT lesions, as compared to B6.Sst1S lesions (Figure 6E,F, Appendix A).

The secondary lesions in both strains were dominated by iNOS+ macrophages (Figure 6E,F, left column). However, there was a greater proportion of uninfected (M.av negative) iNOS+ macrophages in B6 WT large (*p* ≤ 0.0001) and small (*p* = 0.0009) secondary lesions (Figure 6G, Appendix A), while there was a greater proportion of infected (M.av positive) iNOS+ macrophages in B6.Sst1S large and small (*p* ≤ 0.0001) secondary lesions (Figure 6H, Appendix A).

Large B6 WT secondary lesions compared to B6.Sst1S mice were characterized by reduced area of Arg1+ (*p* = 0.0176) and double positive iNOS+/Arg1+ both not overlapping (*p* = 0.0036) and overlapping (*p* = 0.0004) with M.av immunoreactivity (Figure 6E,I–K).

T cell density between B6 WT and B6.Sst1S secondary lesions was also compared. Large secondary lesions were frequently associated with areas of lymphocytic aggregates composed of iNOS−/Arg1−/CD68− cells interpreted as tertiary lymphoid structures containing interspersed CD3ε+ T lymphocytes (Figure 6E,F, right columns). These aggregated lymphoid structures were of similar area in B6 WT and Sst1S large lesions (Figure 6L). Using a tissue classifier, these CD3ε+ segmented cells associated with lymphatic aggregates were removed and analysis was performed only on CD3ε+ cell populations interspersed within large mycobacterial lesions (Figure 6E,F, right columns). T cell density was increased in B6 WT small (*p* = 0.001) (Appendix A) and large (*p* = 0.0025) secondary lesions (Figure 6M) compared to B6.Sst1S.

Taken together, the B6 WT sst1R phenotype was associated with reduced average lesion size, reduced bacterial loads, lower proportion of Arg1+ and double positive iNOS+Arg1+ macrophages, and increased T cell density. Combined, these patterns reflect slower progression of secondary mycobacterial lesions in the sst1 resistant animals.

## 3. Discussion

The respiratory infection of the B6.Sst1S mice via intranasal inoculation with the laboratory strain MAC101 resulted in chronic infection that was not resolved by 12 wpi. At 12 wpi, we documented the development of chronic peribronchiolar granulomatous lesions that contained mycobacterial laden macrophages. To increase intra-pulmonary inoculation, we employed a unilateral intrabronchial inoculation of M.av.

Following unilateral intrabronchial inoculation, we observed the formation of chronic granulomatous lesions in the inoculated lung lobe that had a strong predilection for peribronchiolar distribution. The contralateral un-inoculated lung lobes contained random multifocal interstitial M.av-associated granulomatous lesions. Most likely, the multifocal interstitial lesions of the un-inoculated lung lobes represent secondary, or metastatic, granulomatous lesions originating via hematogenous or lymphogenous spread of mycobacteria. The secondary lesions were more discrete and organized in the lung interstitium, with the overall un-inoculated lobe containing reduced area of acid-fast bacilli (AFB), granulomatous pneumonia lesion area, T cell density compared to the inoculated lobes.

Analysis of the discrete interstitial secondary lesions afforded us the ability to better delineate individual lesions and characterize temporal patterns and variation based on lesions size. Pulmonary disease progressed 12–16 wpi with lesions containing increased percentages of AFB, area of granulomatous pneumonia, iNOS+ macrophages, Arg1+ and double positive iNOS+/Arg1+ macrophages, and T cells. iNOS+ macrophages were the dominant infected macrophage phenotype. There was a smaller amount of Arg1+ and double positive iNOS+/Arg1+ infected macrophages. By 21 wpi, disease progression appeared to plateau and/or resolve based on reduction in AFB compared to 16 wpi yet the presence of dense cytoplasmic bacilli and granulomatous lesions. Both 16 and 21 wpi lesions contained increased density of bacterial aggregates and increased histopathology scores in comparison to 12 wpi lesions.

Though lesion size does not necessarily correlate with the temporal age of the lesion, we found that area of lesion was a reliable quantitative variable associated with disease progression. Larger lesions could be the results of small lesion expansion or coalescence of small lesions. The mean lesion area increased temporally during the experiment (12, 16, 21 wpi) and the appearance of infected and uninfected Arg1+ and double positive iNOS+/Arg1+ macrophages were a specific finding associated with larger lesions. Through comparisons with B6 WT mice, we found that B6.Sst1S mice exhibited enhanced susceptibility to M.av infection by 16 wpi with increased mean area of granulomatous lesions in the un-inoculated lobes and increased area of AFB. This is the first study that demonstrated enhanced susceptibility in B6.Sst1S mice to a non-tuberculosis mycobacterium. Moreover, evidence of enhanced susceptibility was most saliently observed among secondary lesions in the un-inoculated lobe, as mean area of granulomatous lesions was elevated in B6.Sst1S un-inoculated but not inoculated lobes compared to B6 WT. The slow-developing susceptibility to M.av in B6.Sst1S at 16 wpi took longer to develop than with M. tb infection, which is apparent between 9–12 wpi in the lungs [31].

Granulomatous lesions in the susceptible phenotype were also associated with increased infected and uninfected Arg1+ and double positive iNOS+/Arg1+ macrophages and reduced T cell density. Taken together, our studies using temporal, spatial, and genetic susceptibility criteria, demonstrated that increased Arg1 expression by macrophages in chronic M.av lesions represents a conserved macrophage biomarker of granulomatous pneumonia progression. The appearance of double positive iNOS+/Arg1+ macrophages demonstrates that a binary M1/M2 macrophage polarity paradigm is insufficient to characterize the in vivo dynamics of macrophage activation within mycobacterial granulomas.

Increased Arg1+ in larger and susceptible granulomatous lesions may suggest a role for Arg1 as a marker of advanced and/or older resolving lesions. Upregulated expression of arginases Arg1 and Arg2 has also been associated with pathological processes ranging from cardiovascular, immune-mediated, neurodegenerative disorders and tumors, suggesting that arginase may serve as a potential biomarker of progression and severity in various tissues [43]. In mycobacterial infections, Arg1+ macrophages were consistently detected in granulomas caused by various mycobacterial species and in various animal hosts. The IHC analysis of human and non-human primate (NHP) TB lesions found macrophages expressing both iNOS and Arg1 enzymes [16,44]. In NHP granulomas, the Arg1+ macrophages were found primarily in the outer layer suggesting their role in limiting lung pathology [16]. In our findings, however, infected and uninfected Arg1+ or double positive iNOS+/Arg1+ macrophages were usually found in central areas of granulomatous lesions. The disease-promoting effects of Arg1+ macrophages in TB can be explained by arginine depletion and, thus, reducing nitric oxide (NO) production, a critical effector of host resistance to M.tb in mice [45].

Unlike M.tb infection, iNOS deficiency is not associated with greater susceptibility to M.av infection in mice [8,46]. Therefore, depletion of arginine and reduced NO are unlikely to cause disease progression. Perhaps, gradual accumulation of the Arg1+ macrophages during the chronic course of the M.av granuloma progression represents an adaptive macrophage reaction to limit lung tissue damage by NO. Furthermore, depletion of L-arginine due to prolonged Arg1 expression has been shown to suppress T cell proliferation and cytokine release [47], suggesting an Arg1 role in mitigating T cell-driven immunopathology [48]. Of note, after excluding T cell tertiary aggregates, we observed reduced T cell density in larger, more advanced, granulomatous lesions in this B6.Sst1S model. However, a causal relationship between this phenomenon and the increase in Arg1 expressing macrophages remains to be established.

The signals that induce and maintain the Arg1 expression in macrophages of mycobacterial granulomas are not well characterized. Canonically, the M2 macrophage polarization and Arg1 expression are driven by Th2 lymphocyte-derived cytokines IL-4 and IL-13 [49]. Mycobacterial infections, however, are known to drive predominantly Th1 responses [50,51]. Alternatively, mycobacteria were shown to induce Arg1 in macrophages directly via TLR—MyD88—CEBPβ signaling. This pathway required no T cells, IL-4 or IL-13, nor STAT6 activation [17]. Because the sst1-susceptible mice expressed higher levels of Arg1 in M.av lesions, we hypothesized that this locus may be involved in macrophage polarization as well. Our previous studies demonstrated that the aberrant macrophage activation underlies the sst1 susceptible phenotype [36,38]. Upon activation, the mutant macrophages upregulate type I IFN (IFN-I) pathway that drives the integrated stress response (ISR) via upregulation of PKR and ATF4 and ATF3 transcription factors [34]. Of note, the promoter for the Arg1 gene includes binding sites for ATF3. Therefore, we wanted to determine whether Arg1 in the B6.Sst1S background can be upregulated via the IFN-I-mediated ISR in a cell autonomous manner. We used B6.Sst1S-YFP-IFNβ knockin reporter mice that utilizes YFP as a surrogate for IFNβ and allows the identification of the IFNβ-producing cells in situ. IFNβ has been identified as a possible correlate of susceptibility in human tuberculosis [52,53,54] and is implicated in sst1-mediated macrophage susceptibility [34,36,37]. By utilizing the fmIHC approach, we investigated co-localization of IFNβ/YFP with Arg1, iNOS, and M.av. However, we did not observe significant co-localization of Arg1 and IFNβ within mycobacterial lesions. In contrast, IFNβ/YFP expression is colocalized with M.av-infected iNOS+ Arg1− histiocytes. Further studies using this reporter mouse will allow characterization of the respective roles of the Arg1+ and IFNβ-producing cells in shaping the mycobacterial granuloma trajectories, by allowing pre-classification of cell types for subsequent downstream single cell analysis using spatial transcriptomics platforms.

### Study Limitations

We have chosen to use a laboratory adapted virulent M.av strain MAC 101 (Chester, ATCC 700898) because this strain displayed a stable and virulent phenotype in other mouse studies: (i) it multiplied in vivo and the infection was not completely resolved, (ii) it induced T cell responses and inflammatory cytokine production, and (iii) promoted inflammatory cell recruitment and pulmonary granulomatous pneumonia [9]. This strain is also a standard strain for the evaluation of drug susceptibility in vitro and in vivo 1. In standard immunocompetent mouse strains B6 and BALB/c the respiratory infection with this strain produces progressive infection with plateauing and declining CFU counts within 8–12 wpi [23]. Though our data demonstrated that B6.Sst1S mice were more susceptible to M.av infection than B6 WT, unlike in M.tb infection [31], we found no caseating or organized necrotic granulomas in the lungs of the B6.Sst1S mice intranasally inoculated at 12 wpi or intrabronchially inoculated between 12–21 wpi. The chronic infection established in our experiment differs from other reports using more virulent M.av strains in the commonly used C57BL/6 mouse strain which have observed necrotic granulomas [7,55]. In one study, infection with 102 CFU of M.av (strain 25291) resulted in progressive disease and granulomas with central necrosis by 16 wpi [7]. Necrotic granulomas were also observed in a study using aerosol inoculation with a virulent M.av strain in C57BL/6 mice [55]. Therefore, the utilization of B6.Sst1S and clinical isolates of M.av could provide a mouse model that recapitulates more severe forms of human disease.

Although the small number of animals analyzed at each timepoint is a limitation in this study, the digital IA approach enabled the analysis of hundreds of pulmonary lesions. Statistically significant results were observed both temporally, between mouse strains, and among large and small secondary lesions at 16 wpi. Another limitation is a narrow focus on macrophage M1/M2 markers. The integration of T cell markers in spatial analysis will prove informative in linking local macrophage phenotypes with T cell responses within distinct granuloma compartments, as described by Gideon et al. [56].

Technical limitations also include an impeded accuracy of segmentation of histiocytes due to their occasional multinucleation, as well as heterogeneity in cytoplasmic area and shape. This contrasts with the mononuclear CD3ε+ cells, which are uniform in their overall size and shape and were easily segmented using DAPI nuclear staining. Therefore, to obtain quantitative outputs for the macrophage markers iNOS and Arg1, we utilized the Area Quantification (AQ) module to measure positive pixel immunoreactivity. While this provides an accurate representation of the area of each biomarker and pixel overlay between multiple biomarkers, it does not directly phenotype individual cells.

Another limitation of this study is the use of a singular plane for microscopic quantitative evaluation of biomarkers and AFB staining, acknowledging the lung is a three dimensional and heterogeneous environment. Future research would benefit from 3D imaging approaches such as confocal and light sheet microscopy of cleared and immunolabeled tissues. Despite these limitations, our approach provides a semi-automated pipeline and quantitative assessment of NTM lesions that can be implemented on a larger scale in preclinical studies.

In conclusion, the B6.Sst1S mouse model partnered with the showcased methodologies will serve as a useful tool to further understanding mechanisms underlying susceptibility to pulmonary MAC infections in immunocompetent hosts and for cost-effective assessment of therapeutics and vaccines against NTM disease.

## 4. Materials and Methods

### 4.1. Mice and Mycobacterium avium Infection

Sixteen B6.Sst1S mice (10 females and 6 males) were enrolled in the study to assess mycobacterial CFU and pulmonary histopathology following 10^6^ M.av (Strain 101) intranasal inoculation at 4-, 6-, 8-, 10-, 12-weeks post infection (wpi). In addition, B6 WT (*n* = 12) and B6.Sst1S (*n* = 10) female mice were enrolled in a comparison experiment and infected with 10^5^ M.av (strain 101) by left mainstream intrabronchial inoculation. Mice were euthanized at three time points: 12-, 16-, and 21-wpi. One B6 WT mouse and one B6.Sst1S mouse from 12 wpi were excluded from analysis because the lung insufflation failed and lungs were atelectatic. Additionally, one B6.Sst1S 16 wpi sample was excluded because the lungs displayed generalized concurrent eosinophilic crystalline pneumonia, a common background lesion affiliated with this mouse strain. Therefore, for hematoxylin and eosin (H&E) histopathology analysis, the following specimens were used: 12 wpi, *n* = 2 B6.Sst1S and *n* = 3 B6 WT; 16 wpi, *n* = 3 B6.Sst1S and *n* = 4 B6 WT; 21 wpi, *n* = 3 B6.Sst1S and *n* = 4 B6 WT. Additionally, two B6.Sst1S–YFP mice were infected with 10^6^ M.av (strain 101) by left mainstream intrabronchial inoculation and euthanized at 23 wpi. The B6.Sst1S–YFP mice were generated by crossing B6.Sst1S with a reporter mouse that has a Yellow Fluorescent Protein (YFP) reporter inserted downstream of the Ifnb1 gene promoter where YFP serves as a surrogate for Interferon beta (IFNβ) gene expression [57]. Mice were deeply anesthetized intraperitoneally with a mixture of ketamine/xylazine. After euthanasia, the mice were perfused through the retro-orbital sinus with PBS/Heparin using AutoMate in-Vivo Manual Gravity Perfusion System (Braintree Scientific, Braintree, MA, USA) to remove blood from the lungs through the retro-orbital sinus. The retro-orbital sinus was chosen to prevent lung collapse during the perfusion process. These steps were followed by perfusion with 4% paraformaldehyde and lung insufflation with 1% low melting point agarose.

### 4.2. Tissue Inactivation, Processing, and Histopathologic Interpretation

Tissue samples were submersion fixed for 24 h in 4% PFA, processed in a Tissue-Tek VIP-5 automated vacuum infiltration processor (Sakura Finetek USA, Torrance, CA, USA), followed by paraffin embedding with a HistoCore Arcadia paraffin embedding machine (Leica, Wetzlar, Germany) to generate formalin-fixed, paraffin-embedded (FFPE) blocks, which were sectioned to 5 μm, transferred to positively charged slides, deparaffinized in xylene, and dehydrated in graded ethanol. A subset of slides from each sample were stained with H&E for histopathology. Semi-quantitative histopathology analysis was initially conducted by an unblinded single board-certified pathologist (N.A.C.) to characterize the overall heterogeneity and severity of lesions. This knowledge was then utilized to develop a representative ordinal scoring system. Subsequently slides were reanalyzed blinded using the ordinal scoring system (Appendix A). The same semi-quantitative ordinal scoring system was utilized for the intranasal and intrabronchiolar inoculation approaches to assess lung pathology from each specimen at all time points examined (Appendix A).

### 4.3. Fluorescent Multiplex Immunohistochemistry (fmIHC)

Fluorescent mIHC (fmIHC) was performed using Opal fluorophores, which utilize tyramine signaling amplification (TSA) (Akoya Biosciences, Marlborough, MA, USA). One manual 5-plex mIHC panel and one automated 6-plex mIHC panel were developed, optimized, and applied on lung sections from all animals in the study. Staining was performed using antibodies specific for Mycobacterium species (polyclonal), iNOS (M1 macrophage marker), Arg1 (M2 macrophage marker), CD68 (pan-macrophage marker), and CD3ε (T-cells). Optimization details for the manual 5-plex are outlined in Appendix A. Heat induced antigen retrieval (HEIR) was conducted using a BioCare Medical DeCloaking Chamber. Primary antibody protocol parameters including concentration and sequence order were determined through iterative optimization experiments. Next, secondary antibodies conjugated with horseradish peroxidase and TSA conjugated Opal fluorophores were applied for development of primary antibodies. Following development of each fluorophore, HIER was reapplied to strip the previously developed primary-secondary antibody complexes, and the process was repeated in an iterative fashion. After development of the final fluorophore, slides were counterstained with DAPI and mounted with ProLong Gold Antifade Mountant (Thermo Fisher Scientific, Waltham, MA, USA). Isotype controls were evaluated to confirm the absence of nonspecific tissue binding and lung sections from uninfected controls to confirm the specificity of the mycobacterium antibody.

Additionally, a 6-plex, automated mIHC assays for 21 wpi specimens and 23 wpi B6.Sst1S–YFP specimens were performed using a Ventana DISCOVERY Ultra Autostainer (Ventana Medical Systems, Roche Diagnostics, Basel, Switzerland), using DISCOVERY reagents according to the manufacturer’s instructions. Target antigens and protocol details for the automated multiplexes are outlined in Appendix A. The Ventana Discovery Ultra platform was also utilized to generate monoplex diaminobenzidine (DAB) assays utilizing serial sections derived from 23 wpi B6.Sst1S–YFP lungs to target M.av and green fluorescent protein (GFP).

### 4.4. Slide Scanning and Image Analysis

Whole slide images (WSI) of fmIHC slides were digitized at 200× total magnification using the Vectra Polaris Automated Quantitative Pathology Imaging System (Akoya Biosciences). Exposure times were established by applying autoexposures on a region of interest (ROI) on B6.Sst1S slides with copious target proteins ensuring no pixel saturation. The remaining slides were digitized using the same exposure time parameters. An unstained lung section exposed to experimental HIER conditions was also imaged to create an autofluorescence signal that was removed from fmIHC WSI using InForm software 2.4.9 (Akoya Biosciences). Inform Software was also used to computationally spectrally unmix digitized fmIHC slides using a pre-generated spectral library.

Digitized WSI were analyzed using the Halo IA software v3.2 (Indica Labs, Corrales, NM, USA). Each slide was initially manually annotated to remove tissue artifacts (tissue folds, dust, fluorescent precipitates, etc.) and in two slides, focal areas of eosinophilic crystalline pneumonia in the left caudal lung lobe as evidenced by regional over-expression of the Arginase-1 and confirmation of the correlated histologic phenotype by a pathologist (N.A.C.). Next, each slide was manually annotated to denote the left (inoculated) and right (un-inoculated) lung lobes. Threshold values for visualization of each fluorophore were modified to minimize background signal and maximize specificity of target proteins. IA was performed using the Tissue Classifier (TC) (a train-by-example machine learning algorithm utilizing a random forest hierarchical decision tree approach), Area Quantification (AQ), and High-Plex Fl (HP) modules. All module threshold values and parameters were modified and consistent for 12 and 16 wpi specimens, while unique parameters were created for 21 and 23 wpi specimens given different fluorophore intensities. TC was trained to recognize granulomatous lesions containing iNOS+ histiocytes, M.av, and CD3ε+ T cells. A second TC algorithm was designed to specifically recognize areas of T cell aggregates, characterized by abundant DAPI and CD3ε+ staining in the absence of mycobacterium antigen or macrophage markers. Additionally, a third TC algorithm was designed to recognize iNOS+ or Arg1+ histiocytic within lesions, while excluding CD3ε+ and other mononuclear cells.

TC outputs included the total percentage of lung classified based on the total tissue area examined (μm^2^). Lesion specific analysis was conducted in the un-inoculated lung lobes at 12 and 16 wpi using the following criteria defined by the authors. TC granulomas in the un-inoculated lobe were separated into groups based on lesion area (μm^2^). For each unique specimen, lesions equal to and above the 90th percentile by area were annotated as “large secondary lesions,” and lesions between the 50th and 60th% percentile (inclusive) were annotated as “small secondary lesions.” Lesions were excluded from secondary lesion classification if they were directly affiliated with bronchiolar epithelium. Appendix A shows representative annotated lesions.

AQ outputs included the total percentage of tissue with immunoreactivity based on the total tissue area examined (μm^2^), as well as total percent positivity for two or more fluorophores (overlapping or non-overlapping fluorophores). AQ analysis was conducted on the whole lung lobe level and within the annotated un-inoculated lung lobe lesions, as described above. HP outputs rely on cell segmentation via DAPI nuclear segmentation. The HP module was used to segment CD3ε+ cells (T cells) to determine density of T cells (# of CD3ε+ cells/μm^2^). The HP module and T cell density analysis was conducted on the whole lung lobe level, and in secondary lesions that excluded TC classified T cell aggregates.

### 4.5. Mycobacterial Staining and Quantification of Loads in Lung Sections

The brightfield acid-fast staining of *M. avium* infected lung sections was performed using new Fuchsin method (Poly Scientific R and D Corp., cat no. K093, Bay Shore, NY, USA) according to the manufacturer’s instructions.

FFPE M.av infected lung specimens were stained for acid fast bacilli (AFB) using the O-rhodamine B method to visualize mycobacteria as previously described with few modifications [38]. Following deparaffinization and rehydration through graded ethanol washes (100%, 95%, 70%, 50%, 25% and distilled water), the sections were stained with auramine O-rhodamine B at 37 °C for 15 min, excess stain was removed by washing sections in 70% ethanol (3 times for 1 min each). Sections were dehydrated and mounted with a permount mounting medium.

WSI of AFB-stained slides were digitized at 400× total magnification using the Vectra Polaris Automated Quantitative Pathology Imaging System (Akoya Biosciences). The 480-filter cube was used for imaging as fluorescent AFB illustrated the strongest signal to noise in this channel. As described previously, Inform Software 2.4.9 (Akoya Biosciences) was used to computationally spectrally unmix and remove autofluorescence from digitized AFB slides. Digitized WSI were analyzed using the Halo IA software v3.2 (Indica Labs, Corrales, NM, USA) using similar methodology as described previously for fmIHC analysis (removal of tissue artifacts, annotation of lung lobes, generation of threshold visualization values). The Halo AQ tool was used on AFB-stained lung sections to quantify the percentage area of AFB in inoculated and un-inoculated lung lobes.

### 4.6. Statistical Analysis

For ordinal scoring, a Kruskal–Wallis test with Benjamini, Krieger, and Yekutieli correction for multiple comparisons was used given the nonparametric, noncontinuous nature of the data. For fmIHC IA outputs, one-way analysis of variance with post hoc Tukey correction for multiple comparisons and unpaired Student *t*-test were performed. All statistical analyses were performed using GraphPad Prism version 8.4.3 (GraphPad Software, La Jolla, CA, USA) and, for all tests, *p* < 0.05 was considered statistically significant.

## Figures and Tables

**Figure 1 ijms-23-05999-f001:**
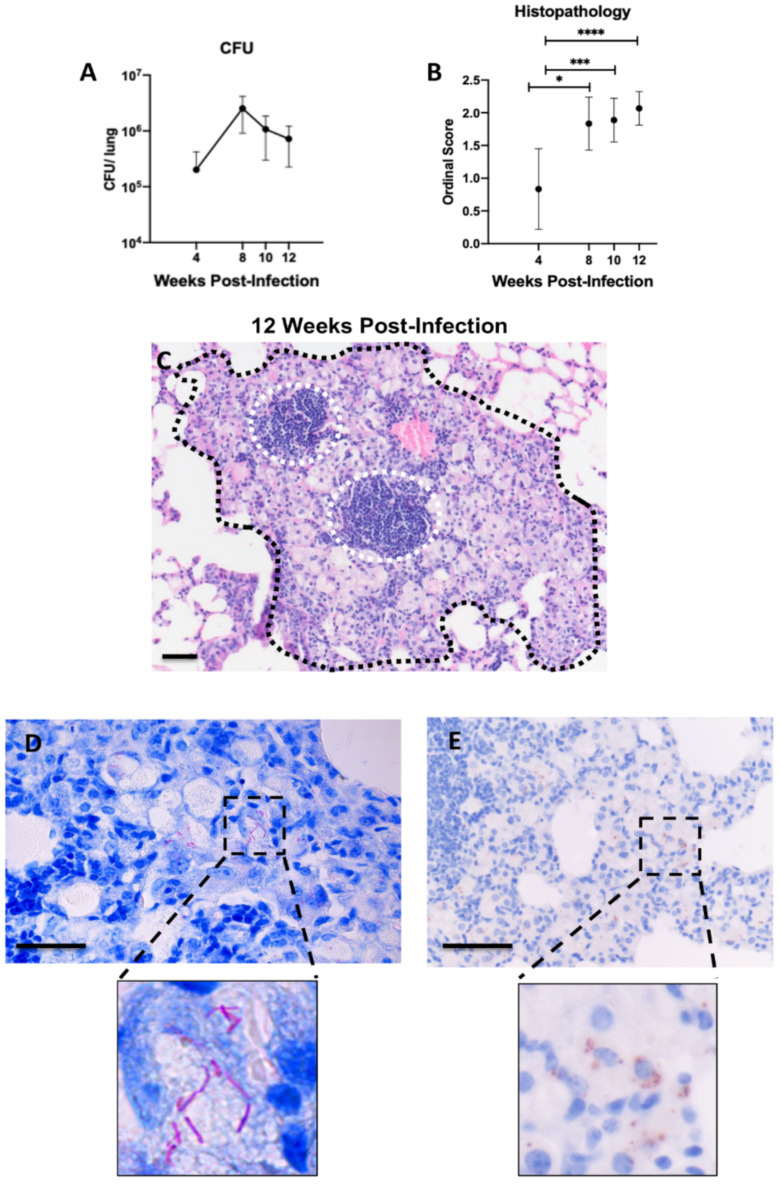
Establishment of chronic pulmonary *M. avium* infection with intranasal inoculation 4–12 weeks post-infection (wpi) in B6.Sst1S mice. (**A**) CFU/lung following intranasal (IN) 10^6^ inoculation; (**B**) histopathology scores of 10^6^ CFU IN *M. avium* inoculated mice at 4, 8, 10, and 12 wpi; (**C**) representative histological micrograph of granulomatous lesion (black hash) with intralesional lymphoid aggregates (white hashed circles) caused by *M. avium* at 12 wpi assessed by hematoxylin and eosin (H&E) and visualized using brightfield microscopy; (**D**) detection of *M. avium* Acid-fast bacilli using Ziehl-Neelsen stain and visualized using brightfield microscopy; (**E**) diaminobenzidine (DAB) immunohistochemistry detection of Mycobacterial antigen visualized using brightfield microscopy. High magnification insets are located below (**D**,**E**) to highlight distinct differences in these modalities utilized to visualize mycobacterium bacilli and antigen respectively. For (**A**), each data point represents average CFU/lung; *n* = 6 (4 wpi), *n* = 2 (8 wpi), *n* = 3 (10, 12 wpi). For (**B**), each data point represents the mean from a single tissue section; *n* = 18 (4 wpi), *n* = 6 (8 wpi), *n* = 9 (10 wpi), *n* = 15 (12 wpi). Data are expressed as means ± SD. * *p* < 0.05, *** *p* < 0.0005, and **** *p* < 0.00005. Original magnification, 100× (**C**) and 200× (**D**,**E**); scale bars = 50 μm.

**Figure 2 ijms-23-05999-f002:**
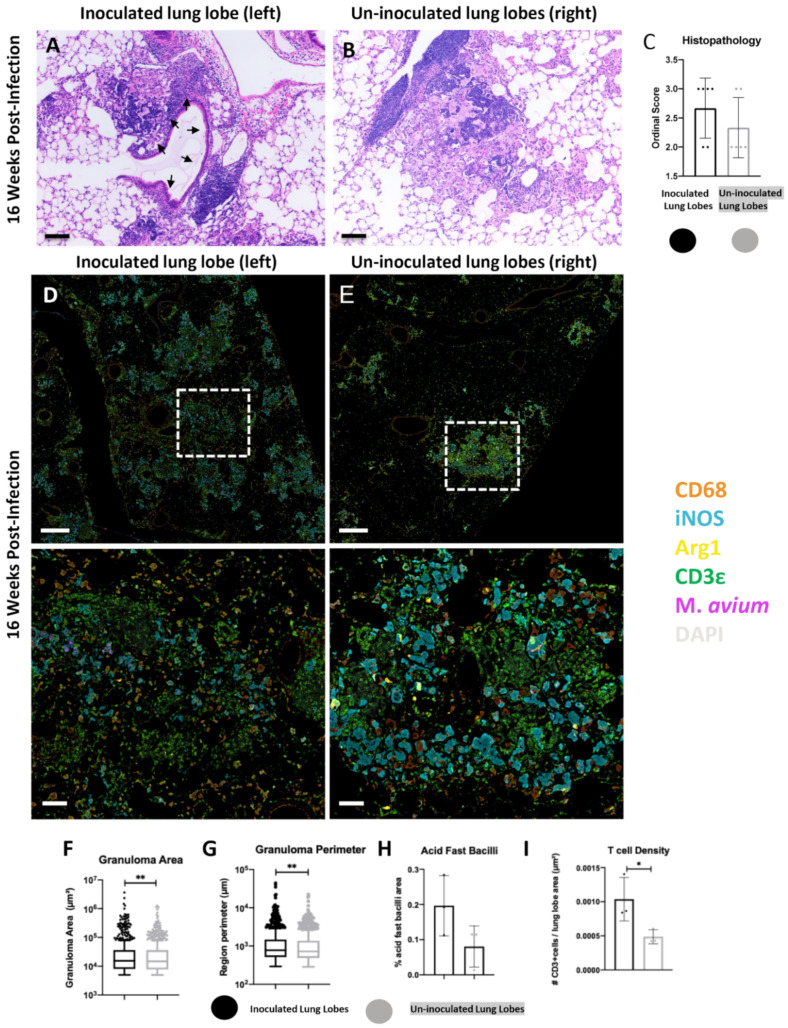
Comparison of inoculated (**left**) and un-inoculated lung lobe (**right**) lesions at 16 weeks-post infection (wpi) following unilateral left mainstream intrabronchial *M. avium* inoculation in B6.Sst1S mice. (**A**) representative micrograph of primary lesions in the inoculated left lung lobe with granulomatous infiltrate in immediate proximity to a bronchiole (black arrows); (**B**) representative micrograph of secondary lesions in an un-inoculated right lung lobe composed predominantly of granulomatous infiltrate with intralesional lymphoid aggregates. (**A**,**B**) assessed by hematoxylin and eosin (H&E) and visualized using brightfield microscopy; (**C**) ordinal histopathology scores, inoculated vs. un-inoculated lung lobes; (**D**,**E**) representative raw fluorescent multiplex immunohistochemistry (fmIHC)images of inoculated (**left**-**D**) and un-inoculated (**right**-**E**) lung lobes. Lesions in the left inoculated lobe coalesced and extended outwards from airways, while lesions in uninoculated lung lobes were randomly distributed in the interstitium. The lower row in (**D**,**E**) represents higher magnification images of the white hashed boxes outlined in the top row; (**F**) granuloma lesion area calculated using random forest tissue classification; (**G**) granuloma lesion perimeter calculated using random forest tissue classification; (**H**) Acid Fast Bacilli staining area quantification (AQ), and (**I**) T cell density. For (**C**), each data point represents a single tissue section, *n* = 6. For (**F**,**G**), each data point represents a single lesion; *n* = 662 (inoculated lobes), *n* = 913 (uninoculated lobes); lesion area and perimeter were calculated using random forest tissue classification. For (**H**,**I**), each data point represents a single animal; *n* = 3 (inoculated lobes), *n* = 3 (uninoculated lobes). Data are expressed as means ± SD or with Tukey whiskers. Asterisks denote *p*-values with a Mann–Whitney test (*) or Student’s unpaired *t*-test (*-*). * *p* < 0.05, ** *p* < 0.005. Original magnification, 100× (**A**,**B**, **D**,**E**, bottom row) and 20× (**D**,**E**, top row)100×. Scale bars; 100 μm (**A**,**B**,**D**,**E**, bottom row); 500 μm (**D**,**E**, top row).

**Figure 3 ijms-23-05999-f003:**
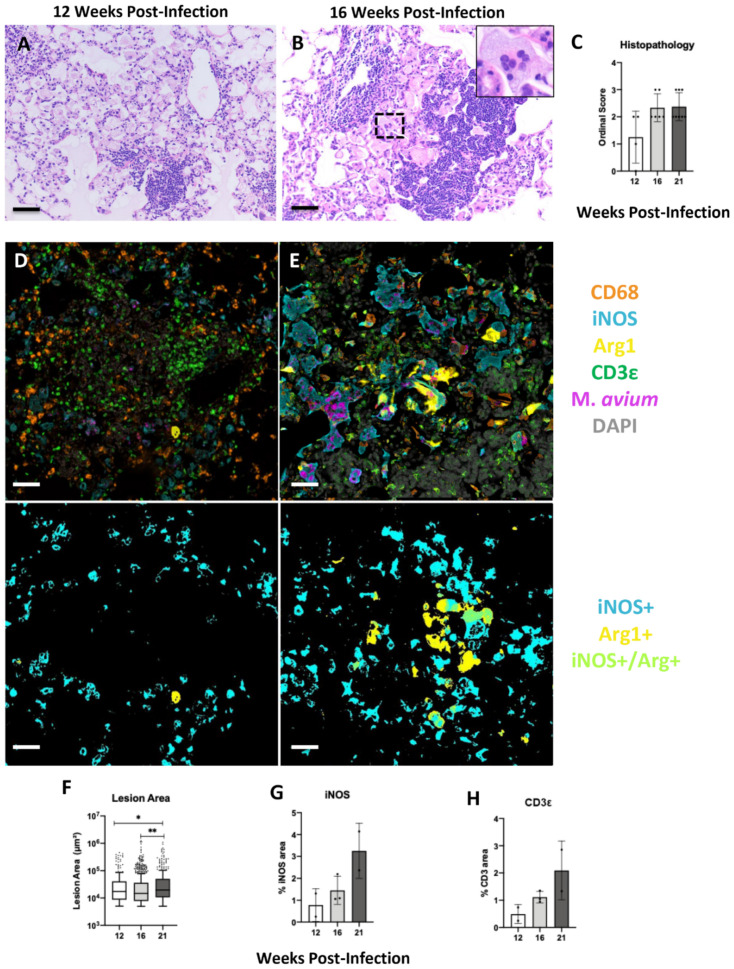
Progression of secondary lesions between 12-, 16-, and 21-weeks-post infection (wpi) in un-inoculated right lung lobes following unilateral left mainstream intrabronchial *M. avium* inoculation in B6.Sst1S mice. (**A**) representative histological micrograph of early secondary lesion at 12 wpi assessed by hematoxylin and eosin (H&E) and visualized using brightfield microscopy; (**B**) representative histological micrograph of a large secondary lesion at 16 wpi containing abundant plump reactive macrophages and multinucleated giant cells (black hashed box and inset) assessed by H&E and visualized using brightfield microscopy; (**C**) ordinal scores, un-inoculated lung lobes at 12, 16, and 21 wpi; (**D**,**E**) representative raw fluorescent multiplex immunohistochemical (fmIHC) image of secondary lesion at 12 wpi (**D**-**upper left**), and 16 wpi (**E**-**upper right**), with corresponding Area quantification (AQ) overlays of iNOS+, Arg1+, and iNOS+/Arg1+ (lower panels) visualized with Halo Image Analysis software v3.2 (Indica Labs, Corrales, NM); (**F**–**H**) un-inoculated lung lobes at 12, 16, and 21 wpi; (**F**) granuloma lesion area calculated using random forest tissue classification; (**G**) iNOS AQ, H. CD3ε+ AQ. For (**C**), each data point represents one tissue section; *n* = 4 (12 wpi), *n* = 6 (16 wpi), *n* = 8 (21 wpi). For (**F**), each data point represents one lesion; *n* = 265 (12 wpi), *n* = 913 (16 wpi), *n* = 306 (21 wpi). For (**G**,**H**), each data point represents a single animal; *n* = 2 (12–21 wpi), *n* = 3 (16 wpi). Data are expressed as means ± SD or with Tukey whiskers. Asterisks denote *p*-values with the Kruskal–Wallis test (*) or one-way ANOVA (*-*). * *p* < 0.05, ** *p* < 0.005, *** *p* < 0.0005. Original magnification 100× (**A**,**B**,**D**,**E**). Scale bars: 50 μm (**A**,**B**,**D**,**E**).

**Figure 4 ijms-23-05999-f004:**
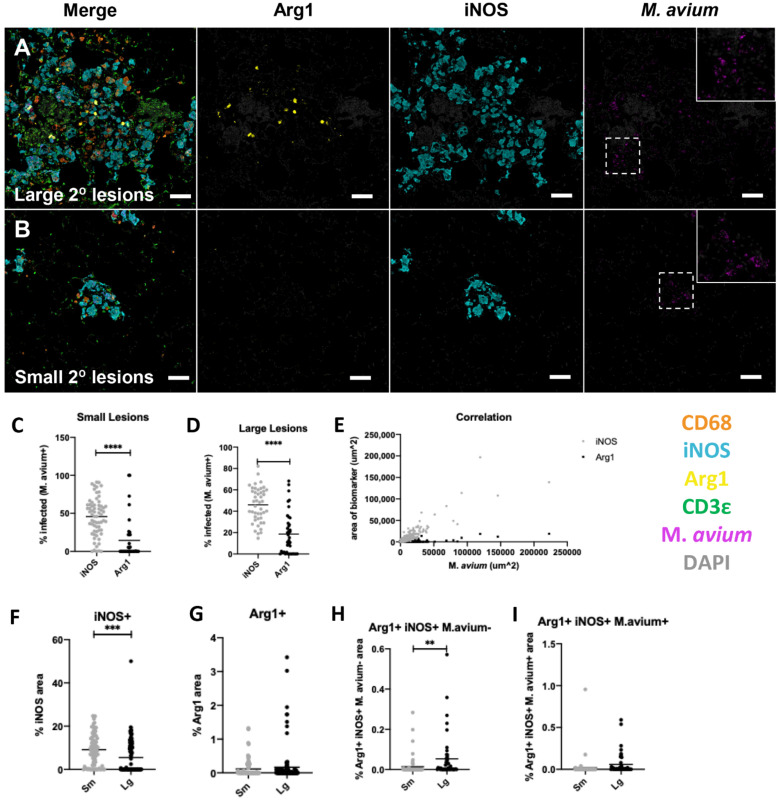
Characterization of macrophage populations in large and small secondary lesions at 16 weeks post-infection (wpi) following unilateral left mainstream intrabronchial *M. avium* inoculation in B6.Sst1S mice. (**A**,**B**) representative raw fluorescent multiplex immunohistochemical (fmIHC) image of large secondary lesions (**A**-**top row**), and small secondary lesion (**B**-**lower row**), with all filters turned on, or only Arg1, iNOS, or *M. avium* individually (**C**) percentage antigen overlay of *M. avium*+ area with iNOS+ and Arg1+ in small lesions; (**D**) percentage antigen overlay of *M. avium*+ area with iNOS+ and Arg1+ in large lesions; (**E**) correlation between *M. avium* area on *x*-axis and iNOS or Arg1 area on *y*-axis; (**F**–**I**) AQ of iNOS, Arg1, and *M. avium* overlay in small and large granulomatous lesions; (**F**) iNOS AQ; (**G**) Arg1 AQ; (**H**) Arg1+/iNOS+/*M. avium*− AQ; (**I**) Arg1+/iNOS+/*M. avium*+ AQ. For (**C**–**I**), each data point represents a single lesion. For (**C**), *n* = 66 (iNOS), *n* = 36 (Arg1); for (**D**), *n* = 45 (iNOS), *n* = 40 (Arg1); for (**E**), *n* = 883; for (**F**–**I**), *n* = 67 (large lesions), *n* = 45 (small lesions). Data are expressed on scatter dot plots with medians. Asterisks denote *p*-values with Student’s unpaired *t*-test. ** *p* < 0.005, *** *p* < 0.0005, and **** *p* < 0.00005. Original magnification 100 × 200 × 100 × (**A**,**B**). Scale bars: 50 μm (**A**,**B**).

**Figure 5 ijms-23-05999-f005:**
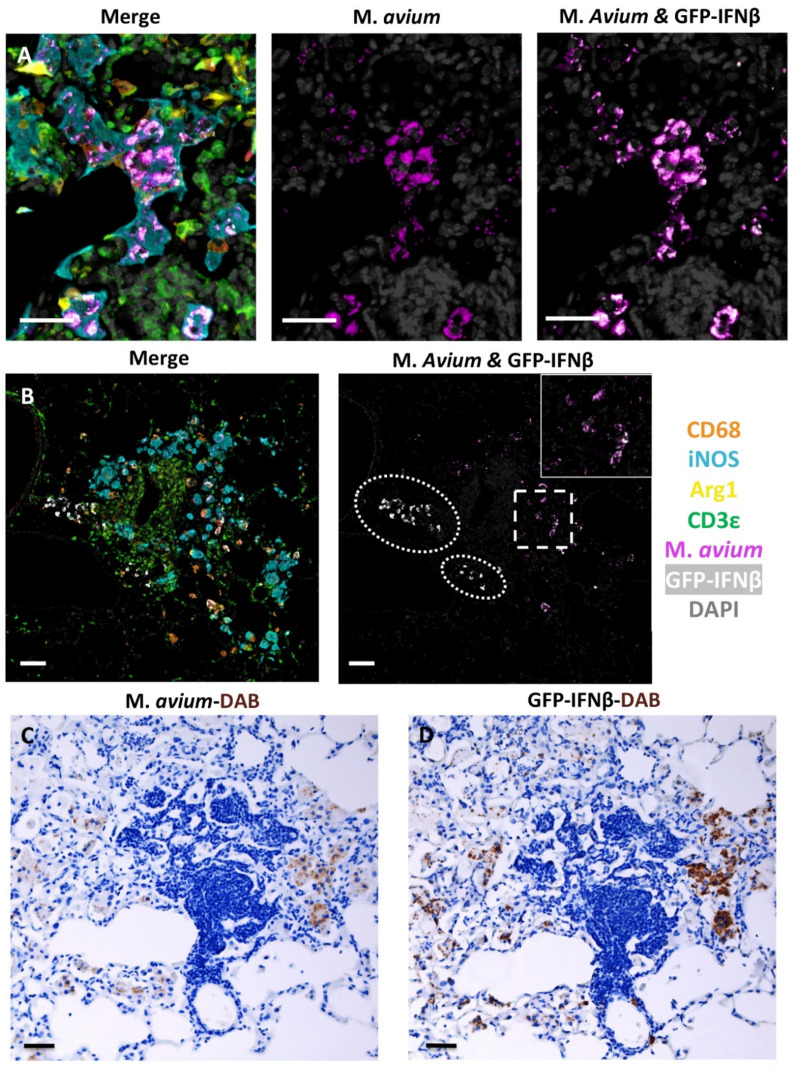
GFP-IFNβ expression in B6.Sst1S.YFP-IFNß mice intrabronchially inoculated with *M. avium* at 23 weeks post-infection (wpi). (**A**,**B**) representative raw fluorescent multiplex immunohistochemistry (fmIHC) images. All filters on (**left**-**A**,**B**), *M. avium* only (**middle**-**A** only) or *M. avium* and GFP-IFNβ (**right**-**A**,**B**). (**A**) GFP-IFNβ predominantly localized with iNOS+ *M. avium*+ macrophages; (**B**) GFP-IFNβ was also expressed in iNOS−/Arg−/CD3ε− mononuclear infiltrate (white hashes); (**C**) *M. avium*− Diaminobenzidine (DAB) IHC.D: GFP-IFNß DAB IHC stain. (**C**,**D**) are serial sections illustrating GFP-IFNß expression predominates in macrophages with *M. avium*. (**C**,**D**) visualized with brightfield microscopy. Original magnifications 200× (**A**) or 100× (**B**,**C**) 400×. Scale bars: 50 μm (**A**–**C**).

**Figure 6 ijms-23-05999-f006:**
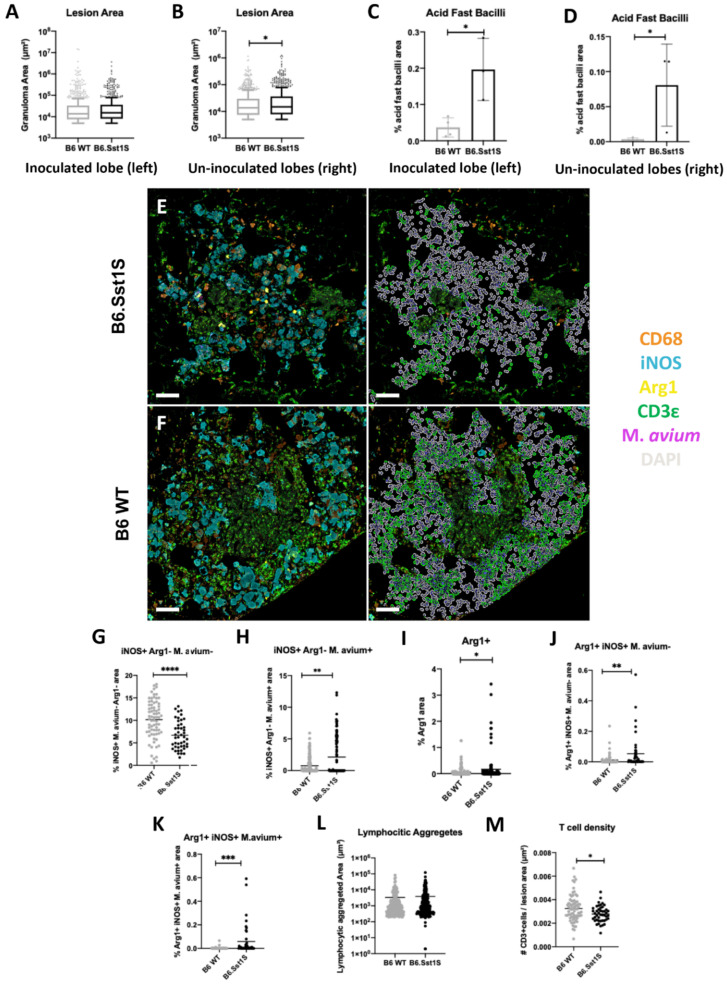
Comparison of immunopathology in inoculated (**left**) and un-inoculated lung lobes (**right**) from *M. avium* inoculated B6 WT compared to B6.Sst1S at 16 weeks-post infection (wpi) and the impact of sst1-S in large secondary lesions at 16 wpi. (**A**–**D**) inoculated vs. un-inoculated lobes; (**A**) Inoculated lobe lesion area; (**B**) un-inoculated lobe lesion area; (**C**) inoculated lobe Acid Fast Bacilli (AFB) area quantification (AQ); (**D**) un-inoculated lobe AFB AQ; (**E**,**F**) representative raw fluorescent multiplex immunohistochemistry (fmIHC) image of B6.Sst1S (**upper left**-**E**) and B6 WT (**lower left**-**F**) large secondary lesions. Right images in (**E**,**F**) are high-plex phenotyping overlays with CD3ε cells in green. Note DAPI aggregates were excluded from analysis; (**G**–**K**) iNOS, Arg1, and *M. avium* AQ immunoreactivity in B6 WT vs. B6Sst1S large secondary lesions; (**G**) iNOS+/Arg1−/*M. avium*− AQ; (**H**) iNOS+/Arg1−/*M. avium*+ AQ; (**I**) Arg1+ AQ; (**J**) Arg1+/iNOS+/*M. avium*− AQ; (**K**) Arg1+/iNOS+/*M. avium*+ AQ; (**L**) Area of lymphocytic aggregates found in large secondary lesions of B6 WT and B6.Sst1S secondary lesions. Random forest TC was used to classify areas of lymphocytic aggregates (high density DAPI and CD3ε+ staining) within large secondary lesions; (**M**) T cell lesion density in B6 WT and B6.Sst1S large secondary lesions excluding lymphocytic aggregates. For (**C**,**D**), each data point represents a single animal; *n* = 4 (B6 WT), *n*=3 (B6.Sst1S). For (**A**,**B**,**F**–**J**,**L**,**M**), each data point represents a single lesion. For A, *n* = 754 (B6 WT), *n* = 662 (B6.Sst1S); For (**C**), *n* = 984 (B6 WT), *n* = 913 (B6.Sst1S); for (**F**–**J**), *n* = 141 (B6 WT), *n* = 117 (B6.Sst1S); for (**L**), *n* = 225 (B6 WT), *n* = 247 (B6.Sst1S); For M, *n* = 78 (B6 WT), *n* = 48 (B6.Sst1S). Data are expressed on scatter dot plots with medians. Asterisks denote *p* values with Student’s unpaired *t*-test. * *p* < 0.05, ** *p* < 0.005, *** *p* < 0.0005, and **** *p* < 0.00005. Original magnification 100× (**E**,**F**). Scale bars: 50 μm (**E**,**F**).

## Data Availability

Data are contained within the article or Appendix A. Additional raw data are available from the corresponding authors upon reasonable request.

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
