# Peer review of "Progression and Dissemination of Pulmonary Mycobacterium Avium Infection in a Susceptible Immunocompetent Mouse Model"

_ijms, 2022, doi:10.3390/ijms23115999_

Round 1

Reviewer 1 Report

In this manuscript authors used classical histological and morphological evaluation to characterize Mycobacterium avium (MAC) granulomatous inflammation in mice. As we know, there are very few suitable mice model to study human clinical phenotype this study has a lot of medical significance. However, my major concern is image quality and markings to make it readable for wider audience. For example,

Fig1 granulomatous structures

Fig 2 BALT, Pathologies and co-localization for fluorescent images

Fig 3 Alveolar macrophages and multinucleated macrophages

Fig 4/5/6 Co- localization of iNOS+ /Arg1+ / M. Avium should be magnified and marked

Antibodies used for tagging Mycobacterium sp. should be mentioned

Minor,

Proofreading for font and format is required

Superscript should be used instead of ^

CD3e should be CD3ε

Reviewer 2 Report

The followings are my comments. If it is useful for the author, please provide them.

General comment: Since human infection by NTM continues to exist as the most difficult problem to treat, it is significant to develop an experimental animal model that can be extrapolated to human treatment.

L-16: Please write the presumed reason why this infection is increasing simply.

L17-18: That's exactly right.

L37~: This introduction provides the necessary information so that even those who are not familiar with NTM infection can understand it well.

However, since I have written too much about histopathology, immunopathology, mouse model needs, etc., the author should organize the introduction focusing on the most important points.

L129: Is this title the title of this paper itself? Isn't the "histopathological feature of lung lesions due to nasal inoculation" better?

L154-156: This description may be done in the discussion.

L156-157: Please write a comparison with M. tb infection in the discussion. Please write only that caseous degeneration was not seen in this experimental example.

L169: Magnification of histopathological photographs C, D, and E should be indicated by a rod indicating micron.

L170-171: Isn't this title better "histopathological features of lung lesions due to unilateral intrabronchial inoculation" as it corresponds to 2.1?

L172-174: This is better to write in the discussion.

L174-176: It would be better to write this in the discussion as well.

 It is better to objectively describe only what the observations were in the Results.

L182-186: It is better to write only the characteristics and distribution of cell infiltration in these Results, and write in the Discussion about the possibility of histopathological development.

L182-186: The findings of nasal inoculation should be written in 2.1. The differences between the two inoculation methods should be summarized in the discussion.

L187-189: Write in the materials and methods that a comparison of histological findings over time should be made on both inoculation routes.

191-192: This line is discussion.

L192-195: Please write these lines in the Discussion.

L196-198: Selection of these markers should be written in M&M. 

L200-204: These lines should be in M&M.

L261-262: Magnification of histopathological photographs A and B should be indicated by a rod indicating micron.

L263: Progression of secondary lesions in the un-inoculated lobes after intrabronchial inoculation?

L318-319: Magnification of histopathological photographs A and B should be indicated by a rod indicating micron.

L362-366: Please write these lines in the Discussion.

L405~: Magnification of histopathological photographs B-E should be indicated by rod indicating micron.

L443: “As previously discussed”  It is better to organize the result and discussion separately.

L460-462: This should also be moved to the discussion.

L487~: I think this discussion is a bit verbose. Select the important results from this experiment, list them individually in order of increasing importance, clarify their significance, and compare them with known reports.

L488-495: This sentence should be written in the introduction. Discussion should begin with a discussion of the most important and novel findings obtained in this study.

Minimize what you wrote in the intro to the discussion.

L488-505 should be in the introduction.

L496-499: This sentence should also be written in the introduction.

L506-520:  In short, briefly describe at the beginning of the Discussion what are the most important results and their significance in this study.

L518~: You need any discussion relating to the difference. Just observed data is not enough.

L518-521: Is this sufficient for the similarity between this experimental model and pulmonary NTM infections? Also, does it mean that the results of experimental infections performed in this model can be extrapolated to human NTM countermeasure research?

L550-561:The association between the immunostaining results obtained in this experiment and this consideration is inadequate. It looks like a short review. The author should write what is new about the results of this experiment, even though there are many unclear points in the pathogenesis of acid-fast bacillus infection.

L563-565: I think this part of the discussion is good.

L571-572How can we prove this causal relationship?

L590-591: Why is this?

L638~: It is necessary to enter the approval number of the Animal Care and Use Committee for this animal experiment in M&M.

L639-663: OK

L663-675: OK

L676-701: OK

L676-701: OK

L749-769: OK

L770-776: OK

L799-800: This information should be included in the M&M.

Reviewer 3 Report

I read with great interest the paper. Authors wrote on an important issue non tubercular micobactery are very dangerous and their quality of life and treatment outcome are scarce

Below my suggestions

  1. Introduction: well wrote. Add if you can more information about 

    Nramp1 allele.

    . Linee 47 Acquired Immune Deficiency Syndrome add AIDS
  2. Progression of secondary lesions: explain better why quantitative analysis of all secondary lesions revealed differential expression patterns of iNOS and Arg1 expressing macrophages.
  3. Figure are very excellent
  4. Discussion: you done a very deeply discussion and also appreciate the limitation section
  5. Conclusion: reinforce your excellent take home message
  6. Methods: well done
